# The Inhibitory Effect of Early Pregnancy Factor on Red Meat Neu5Gc-Mediated Antibody Production in CMAH^−/−^ Mice

**DOI:** 10.3390/nu16060905

**Published:** 2024-03-21

**Authors:** Cong Wang, Honglin Ren, Han Wang, Haosong Li, Jian Guo, Yiran Xiao, Yuxi Guo, Mengdi Liu, Fuchun Duan, Pan Hu, Yansong Li, Zengshan Liu, Shiying Lu

**Affiliations:** State Key Laboratory for Diagnosis and Treatment of Severe Zoonotic Infectious Diseases, Key Laboratory for Zoonosis Research of the Ministry of Education, Institute of Zoonosis, College of Veterinary Medicine, Jilin University, Changchun 130062, China; congwang20@mails.jlu.edu.cn (C.W.); renhl@jlu.edu.cn (H.R.); wang_h19@mails.jlu.edu.cn (H.W.); lihs22@mails.jlu.edu.cn (H.L.); jianguo20@mails.jlu.edu.cn (J.G.); xiaoyr20@mails.jlu.edu.cn (Y.X.); yxguo21@mails.jlu.edu.cn (Y.G.); liumd22@mails.jlu.edu.cn (M.L.); duanfc22@mails.jlu.edu.cn (F.D.); hupan@jlu.edu.cn (P.H.); lys@jlu.edu.cn (Y.L.); zengshan@jlu.edu.cn (Z.L.)

**Keywords:** early pregnancy factor, EPF, *N*-glycolylneuraminic acid, Neu5Gc, red meat, CMAH

## Abstract

The meat derived from mammals such as cows, sheep, and pigs is commonly referred to as red meat. Recent studies have shown that consuming red meat can activate the immune system, produce antibodies, and subsequently develop into tumors and cancer. This is due to the presence of a potential carcinogenic compound in red meat called *N*-ethanol neuraminic acid (Neu5Gc). Neu5Gc is a common sialic monosaccharide in mammals, synthesized from *N*-acetylneuraminic acid (Neu5Ac) in the body and typically present in most mammals. However, due to the lack of the CMAH gene encoding the cytidine 5′-monophosphate Neu5Ac hydroxylase, humans are unable to synthesize Neu5Gc. Compared to primates such as mice or chimpanzees, the specific loss of Neu5Gc expression in humans is attributed to fixed genome mutations in CMAH. Although Neu5Gc cannot be produced, it can be introduced from specific dietary sources such as red meat and milk, so it is necessary to use mice or chimpanzees that knock out the CMAH gene instead of humans as experimental models. Further research has shown that early pregnancy factor (EPF) has the ability to regulate CD4^+^T cell-dependent immune responses. In this study, we established a simulated human animal model using C57/BL6 mice with CMAH gene knockout and analyzed the inhibitory effect of EPF on red meat Neu5Gc-induced CMAH^−/−^ C57/BL6 mouse antibody production and chronic inflammation development. The results showed that the intervention of EPF reduced slow weight gain and shortened colon length in mice. In addition, EPF treatment significantly reduced the levels of anti Neu5Gc antibodies in the body, as well as the inflammatory factors IL-6 and IL-1β, TNF-α and the activity of MPO. In addition, it also alleviated damage to liver and intestinal tissues and reduced the content of CD4 cells and the expression of B cell activation molecules CD80 and CD86 in mice. In summary, EPF effectively inhibited Neu5Gc-induced antibody production, reduced inflammation levels in mice, and alleviated Neu5Gc-induced inflammation. This will provide a new re-search concept and potential approach for developing immunosuppressants to address safety issues related to long-term consumption of red meat.

## 1. Introduction

The World Health Organization issued a statement in October 2015, classifying processed meat products such as bacon, ham, and sausages as Group 1 carcinogens (i.e., substances that are known to cause cancer in humans), and red meat such as cattle, sheep, and pork as Group 2A carcinogens (i.e., substances that probably cause cancer in humans). It is estimated that approximately 50,000 individuals worldwide succumb to cancer annually, with the scientific literature suggesting a likely association with the high consumption of red meat. *N*-Glycolylneuraminic Acid (Neu5Gc), a recently identified carcinogenic compound found in red meat, has garnered significant attention in recent years [1]. The salivary acid component Neu5Gc is predominantly located at the termini of cell surface glycoconjugates in mammals, and it plays a crucial role in cellular recognition, adhesion, inflammatory response, as well as tumor cell growth and metastasis in vivo. Under normal circumstances, Neu5Gc is synthesized from Neu5Ac through the catalytic action of CMP-Neu5Ac hydroxylase (CMAH), which facilitates the conversion of Neu5Ac into Neu5Gc [2,3]. During human evolution, the absence of a fragment within the CMAH gene resulted in the inactivation of the hydroxylase responsible for synthesizing Neu5Gc from Neu5Ac, causing humans to lose the production of Neu5Gc. On the contrary, chickens completely lack the CMAH gene, resulting in a normal lack of Neu5Gc in both humans and chickens. However, other mammals, such as pigs, cows, and sheep, express Neu5Gc in their cells [4,5,6]. The human body obtains exogenous Neu5Gc by consuming substances containing Neu5Gc, such as red meat and milk. The sialic acid biosynthesis enzyme in the human body lacks the ability to distinguish between Neu5Gc and Neu5Ac, thus exploiting exogenous Neu5Gc for salivary secretion in Hu cells through this metabolic loophole. Although human sialic acid biosynthesis enzymes do not clearly distinguish between Neu5Ac and Neu5Gc, they are re-encoded by the human immune system, leading to the production of specific antibodies and chronic inflammation. Therefore, this process increases the risk of developing tumors and other diseases [7]. The presence of Neu5Gc has been demonstrated to be significantly elevated in the cell membranes of malignant tumors associated with various organ diseases, including liver, stomach, lung, breast, pancreatic, lymphatic and colon cancers. This further underscores the substantial risk that Neu5Gc poses to human health [8,9,10,11].

Early pregnancy factor (EPF), as the first protein to appear in the serum of pregnant rats, was detected by Morton [12]. The presence of EPF in pregnancy serum hinders T-lymphocyte production by suppressing its mediated immunity, thereby preventing the rejection of the fetus as an alien entity within the mother’s body [13]. The results of skin grafting experiments revealed that the injection of srEPF into the grafts significantly extended the survival time of the transplanted skin, indicating that EPF possesses immunosuppressive properties and effectively mitigates immune rejection induced by skin grafting [14,15]. The EPF exhibits promising potential in the treatment of arthritis, organ transplantation, and burns by effectively suppressing cell-mediated immune responses in vivo.

Although Neu5Gc is not naturally present in the human body, it can be introduced into the body’s tissues through dietary intake and consumption of animal-derived products. Consequently, Neu5Gc can directly engage with the human immune system, thereby triggering an immune response that leads to the production of specific anti-Neu5Gc antibodies. Prolonged interaction between these antibodies and antigens results in chronic inflammation, consequently elevating the susceptibility to diseases such as tumors [16,17]. If we can artificially intervene to inhibit this critical step in antibody production, it is anticipated that the occurrence of inflammation and the potential for cancer development can be mitigated. In this study, we analyzed the inhibitory effect of EPF on antibody production and chronic inflammation development induced by red meat Neu5Gc in CMAH^−/−^ C57/BL6 mice. A 92 bp mouse model with the knockout of CMAH gene exon 6 was used as a simulated human experimental model. Based on the reference to previous studies, we used CMAH^+/+^ mouse spleen cells containing Neu5Gc as a substitute and studied them through feeding and immunization [18,19]. The results showed that EPF effectively inhibited Neu5Gc-induced antibody production, reduced inflammation levels in mice, and alleviated Neu5Gc-induced inflammation. This will provide a novel research concept and an effective solution for developing potential immunosuppressants to address safety concerns associated with the long-term consumption of red meat.

## 2. Materials and Methods

### 2.1. Animals

The CMAH^−/−^ C57/BL6 mouse, with a 92 bp knockout on exon 6 of the CMAH gene, was custom-made by Saiye Biotechnology Co., Ltd., Tokyo, Japan. The breeding cage maintains a 1:1 male-to-female sex ratio for long-term cohabitation and undergoes regular inspections. Upon identification of pregnant female mice with enlarged stomachs, they are separated and housed in individual cages until giving birth. The practice commonly involves housing mice in separate cages approximately 21 days after birth to prevent overcrowding in the breeding area, thereby ensuring compliance with animal welfare regulations. On the contrary, female mice commence follicle development at 20 days of age and attain maturity by 30 days, enabling them to conceive and produce offspring. Neglecting timely separation can lead to breeding disorder and indistinct generations. Subsequently, a total of 30 mice aged between 6 and 8 weeks were successfully bred and housed in individual isolation cages with ad libitum access to food and water on a daily basis. Once the appropriate age for experimentation was reached, an animal model was established. The animal experiments were conducted under the supervision of the Experimental Animal Welfare Ethics Committee at Jilin University, while the Experimental Animal Center at Jilin University breeds and maintains animals in compliance with national ethical welfare requirements for experimental animals.

### 2.2. Experimental Grouping

A Neu5Gc-mediated animal inflammatory model and an EPF intervention comparative model were established using CMAH^−/−^ C57/BL6 mice as experimental subjects. The blank group was administered intraperitoneal injections of physiological saline, while the experimental control group received intraperitoneal injections of spleen cells from CMAH^−/−^ C57/BL6 mice lacking Neu5Gc. The inflammation group was treated with intraperitoneal injections of wild-type C57/BL6 mice spleen cells containing Neu5Gc. The inflammation-promoting (pro-inflammation) group was fed a diet rich in red meat containing Neu5Gc based on the inflammation group. In the pre-experiment, the maximum daily intake of mice was tested, and red meat was cooked and fed to the mice. Finally, the EPF intervention group received injections of EPF protein following the inflammation-promoting regimen. In our preliminary experiment, we fed red meat and injected CMAH^−/−^ mice with cells without Neu5Gc. The experimental results showed no difference compared to the blank group. Therefore, in order to retain the single variable of the control group and the Inflammation group, which is the injection of spleen cells with/without Neu5Gc, we chose to simplify the control group by not feeding red meat and injecting spleen cells without Neu5Gc.

The experimental groups comprised six mice each, which were provided with cooked and chopped red meat as their diet. Intraperitoneal injections of spleen cells (10^7^ per injection) and EPF protein (10 μg per injection) were administered once a week and once every two days, respectively. Changes in body weight were regularly recorded, while the mental state of the mice was observed for any indications of hematochezia, weakness, or other conditions. After one month of experimentation, the mice were euthanized by taking blood from the eyeballs. Adequate water was given the day before execution, and pentobarbital was used for anesthesia. Blood samples were collected to isolate serum, the pathological organs were dissected, and measurements of colon length as well as liver, kidney, spleen, and other organ sizes and weights were recorded. Additionally, the samples were stored at −80 °C for subsequent testing. The animal experiment group and implementation plan are shown in Figure 1.

### 2.3. Validation of the In Vitro Activity of EPF Protein via Rosette Inhibition Assay

Our previous study successfully constructed a recombinant expression plasmid incorporating SUMO tags from the coding sequence region of human EPF. Subsequently, through transformation, induced expression, and purification using nickel column affinity chromatography, we obtained a highly expressed srEPF protein, which was utilized as an immunogen for the generation of highly efficient rabbit-derived polyclonal antibodies. The EPF was then employed in the detection of cellular activity via a rosette inhibition assay. The fresh blood of guinea pigs was extracted from the heart and placed in an anticoagulant tube. The sample was then diluted and combined with a separation solution, followed by centrifugation at 500× *g* for 25 min. After centrifugation, the circular milky white lymphocyte layer was carefully aspirated from the centrifuge tube and transferred to a new sterile centrifuge tube for purification and resuspension. Then, 1 drop of cell suspension was extracted, and the trypan blue staining technique was employed to determine the cell count and calculate cell viability. If the survival rate exceeds 95%, Hanks’ solution containing 20% calf serum without Ca and Mg ions should be utilized to adjust the cell concentration to 2 × 10^6^ cells/mL for backup purposes. Subsequently, a volume of 1 mL of blood was collected from the rabbit’s ear vein and promptly injected into an anticoagulant tube. The mixture was inverted and subjected to two washes with Hanks’ solution devoid of Ca and Mg ions at 2500 rpm for 10 min. A suspension of red blood cells was prepared with a final concentration of 1 × 10^8^ cells/mL and utilized within four hours. The control group, EPF group, and EPF + antibody group were established. Then, 0.1 mL of lymphocyte suspension was combined with 0.1 mL of red blood cell suspension and 0.1 mL of calf serum in the control group by gently mixing them together. In the EPF group, enzyme-cut EPF was added on top of the control group to achieve a final concentration of 20 μg/mL. In the EPF + antibody group, the EPF antibody was added to the EPF group and allowed to fully bind at a temperature of 37 °C for 20 min. Then, it was centrifuged at a speed of 900 r/min for 5 min, and the rate of lymphocyte formation was compared. After fixation, a drop of fixed liquid was placed onto a clean glass slide and left to diffuse naturally. Once dry, it was stained using Rui’s Giemsa staining method and a microscopic examination was performed [20,21].

### 2.4. Detection of Serum Neu5Gc Antibody Levels Using the Indirect ELISA Method

Neu5Gc-BSA conjugate was diluted with coating solution to a concentration of 1 μg/mL and incubated at 37 °C for 2 h. After sealing for 1 h, the serum was diluted to 1:100 and added to a microplate. The serum was incubated at 37 °C for 1 h. PBS blank wells were set. The goat anti-mouse enzyme-linked secondary antibodies were diluted with PBS (5000-fold) and used at a volume of 100 μL/well, incubating at 37 °C for 1 h. The TMB substrate was mixed with equal volumes of solutions A and B, adding 100 μL to each well. The reaction proceeded in darkness at 37 °C for 10 min before the addition of the termination solution. The fluorescence values at OD450 nm were read using a multifunctional enzyme marker [22].

### 2.5. Quantification of Inflammatory Cytokines Using an ELISA Kit

The levels of pro-inflammatory cytokines, including TNF-α, IL-1β, and IL-6, in blood samples were quantified using a Biolegend ELISA kit (430904, 432604 and 430904, Biolegend, San Diego, CA, USA). PBS was added to the intestinal tissue for homogenization. Then, the sediment was discarded, and the supernatant was retained for analysis.

### 2.6. Determination of Myeloperoxidase (MPO) Activity

The measurement of myeloperoxidase (MPO) activity serves as an indicator of neutrophil functionality and activation, making it a valuable tool for assessing inflammation within the body. Additionally, following the instructions provided in a reagent kit, the intestinal tissue can be weighed to facilitate this process (A044-1-1, Nanjing Jiancheng Biotechnology Research Institute, Nanjing, China).

### 2.7. Determination of Tissue and Organ Index

After conducting the experiment, the weight of the experimental mice was measured. Subsequently, the liver, kidney, and spleen tissues were rinsed with physiological saline solution and patted dry using filter paper before obtaining their respective weights. The following was used to determine the tissue organ index and calculate it based on the measured body weight and individual organ weights, as outlined below
Organ Index = Organ Weight (mg)/Mouse Body Weight (g)

### 2.8. HE Staining Analysis of Pathological Changes in Chronic Inflammation

After euthanizing the mice, their colon, liver, small intestine, kidney, and lung tissues were fixed in 4% paraformaldehyde for a duration of 24 h. Subsequently, these tissues were stained with hematoxylin and eosin and meticulously examined under a microscope. Image collection was employed to analyze the presence of tissue inflammation [23].

### 2.9. Immunohistochemical Analysis of CD4 Cell Proliferation and Inhibition

The paraffin sections of the spleen were soaked in paraformaldehyde, followed by dehydration using an environmentally friendly dewaxing solution. Subsequently, antigen retrieval was performed as per the manufacturer’s instructions, and a commercially available immunohistochemical hypersensitivity kit (kit 9710, Maxin Biotechnology, Rockville, MD, USA) was employed to block endogenous peroxidase activity with 3% hydrogen peroxide. After blocking, CD4 (D7D2Z) specific antibody (25229S, Cell Signaling Technology, Danvers, MA, USA) was gently applied onto the section. The kit should be utilized for subsequent steps. An enzyme substrate chromogenic agent (DAB-0031, Maxin Biotechnology) was utilized to visualize the color development on the tissue slices. Finally, the nuclei were stained with hematoxylin and sealed with neutral resin.

### 2.10. Determination of CD Molecules Associated with B Cell Activation by RT-PCR

According to the manufacturer’s instructions, RNAiso Plus reagent (Takara Bio Inc., Otsu, Japan) should be utilized for tissue RNA isolation. Subsequently, cDNA synthesis kits (Farmers Life Science, Fort Collins, CO, USA) were employed for RNA transcription. All of the expression levels were normalized to β-actin levels. The sequences of the primers are presented in Table 1.

### 2.11. Statistical Analysis

The experiments were conducted in triplicate. Duplicate data comparison was performed using one-way analysis of variance (ANOVA). A *p*-value of ≤0.05 was considered statistically significant.

## 3. Result

### 3.1. Cell Level Analysis of EPF Protein Activity

By observing the formation of rose wreaths to detect the activity of early pregnancy factors, a lower rate of rose wreath formation indicates higher EPF activity and stronger immunosuppressive function. The data presented in Table 2 demonstrate a significant decrease in the formation rate of the EPF group (19.3%) compared to the control group (37.3%). However, upon the addition of EPF antibodies, there is an increase in wreath formation rate (31.7%), approaching that of the control group. These findings suggest that under in vitro conditions, the expressed EPF protein exhibits remarkable immunosuppressive capabilities and the prepared EPF antibodies possess specific binding affinity towards EPF, thereby reducing competition for lymphocyte surface binding sites and inhibiting EPF activity.

### 3.2. The Effect of EPF on the Colon Length and Body Weight in Mice with Red Meat Neu5Gc-Induced Inflammation

After conducting the experiment, we proceeded with the euthanasia and dissection of the mice to meticulously observe and document alterations in both colon length and fluctuations in body weight. The changes observed in colon length can serve as an indirect indicator of pathological modifications and inflammation levels within the colon tissue of the mice, often accompanied by corresponding variations in body weight. As depicted in Figure 2A,B, when compared to the blank group (approximately 11 cm), there was a significant reduction in colon length among mice subjected to the pro-inflammation group (around 9 cm), indicating evident damage to their colon tissue. However, the administration of EPF via intraperitoneal injection resulted in the restoration of colon length (about 10 cm) among the EPF intervention group mice, thereby ameliorating the condition characterized by shortened colons with statistically significant differences (*p* < 0.001). These findings strongly suggest that EPF possesses potential therapeutic properties capable of alleviating shortened colons associated with inflammation.

The results demonstrated that the average body weight of mice In the pro-inflammation group increased from 22.30 ± 0.02 g at the beginning of the experiment to 24.20 ± 0.03 g at week 4, whereas the EPF intervention group exhibited an increase from 22.30 ± 0.02 g to 24.70 ± 0.04 g, and the control group showed an increase from 22.30 ± 0.03 g to 25.00 ± 0.10 g over time, indicating significant differences with varying degrees of statistical significance observed among these groups. The underlying reason for this disparity may be attributed to a substantial weight gain resulting from red meat consumption in both the pro-inflammation group and EPF intervention group when compared to the blank control group fed with normal mouse food; however, after seven days of feeding red meat, the injection of wild C57/BL6 mouse spleen cells containing Neu5Gc led to elevated levels of Neu5Gc antibody in mice, exacerbating systemic inflammation and subsequently causing slower weight gain. After intervention by EPF, its therapeutic effect mitigated the deceleration in mouse weight gain. The remaining two groups exhibited no significant fluctuations in body weight and maintained a consistent growth trajectory throughout the entire experimental process, as depicted in Figure 2C.

### 3.3. EPF Reduced the Levels of Anti-Neu5Gc Antibodies in Red Meat Neu5GC-Induced Inflammation Mice

Exogenous Neu5Gc is ingested into the body, incorporated into human tissues, and encountered by the immune system, eliciting an immune response and generating antibodies. The persistent interaction between antibodies and antigens triggers chronic inflammation, thereby initiating and promoting the development of cancerous growths and tumors. Consequently, the measurement of anti-Neu5Gc antibody levels in the body plays a pivotal role in disease diagnosis. In general, the severity of the disease in the body is positively correlated with the level of anti-Neu5Gc antibodies in the serum. As depicted in Figure 3, no significant change was observed in serum antibody content after 1 month of Neu5Gc intervention in the inflammatory group. However, under dual stimulation from feeding and immunity, the anti-Neu5Gc antibody level significantly increased (OD value approximately 0.6) in the pro-inflammation group, which differed significantly from that observed in the blank group (OD value approximately 0.18) (*p* < 0.001). After the intervention of EPF in mice with Neu5Gc-induced inflammation, a significant reduction (OD value approximately 0.23) was observed in the content of anti-Neu5Gc antibodies, indicating an improvement in vivo inflammation and demonstrating that EPF intervention effectively decreased the levels of anti-Neu5Gc antibodies in mice.

### 3.4. EPF Attenuated the Content of Inflammatory Factors in the Serum of Inflammatory Mice

The serum levels of inflammatory cytokines serve as a direct reflection of the extent of inflammation within the body. We assessed the dynamic changes in mouse inflammation by measuring the concentrations of inflammatory cytokines in their serum. The above analysis results indicate that in the inflammatory response of mice to red meat Neu5Gc, there was an upregulation of IL-1β (Figure 4A), IL-6 (Figure 4B), and TNF-α (Figure 4C) inflammatory factors, as well as MPO activity (Figure 4D) to varying degrees. However, after EPF intervention, there was a significant reduction in the levels of these inflammatory factors. This suggests that red meat Neu5Gc has a pronounced detrimental effect on the body by inducing inflammation through the production of specific antibodies. Furthermore, EPF can effectively mitigate Neu5Gc-induced inflammation by significantly reducing the expression levels of various cytokines and intervening in its occurrence and development.

### 3.5. EPF Alleviates Histopathological Damage in Mice with Neu5Gc-Induced Inflammation

The inflammatory group mice exhibited no significant changes or differences compared to the blank group, as depicted in Figure 5A. However, the pro-inflammation group mice displayed varying degrees of increase in liver, kidney, and spleen organ indices, which aligns with the observed inflammatory condition in vivo. Following EPF intervention, the EPF intervention group demonstrated changes that tended to resemble those of the normal control group and exhibited significant differences. These findings indirectly suggest that EPF effectively mitigates systemic inflammation induced by red meat Neu5Gc in mice.

The results depicted in Figure 5B demonstrate that the nuclear architecture of liver tissue remains intact in both the blank control group and experimental control group, with a clear distribution of liver cells and no evidence of cellular degeneration or inflammatory changes. Furthermore, no abnormalities were observed in the structure of the liver tissue. In the inflammatory group, a small number of neutrophils and lymphocytes infiltrate without any signs of liver cell edema or degeneration. Conversely, the pro-inflammation group exhibits evident lesions characterized by focal hepatitis, partial liver cell degeneration and necrosis, increased infiltration of inflammatory cells, as well as severe systemic inflammation. Following EPF treatment, there is a gradual reduction in necrotic cells accompanied by an improvement in inflammatory infiltration.

The control group exhibited an intact intestinal wall, relatively preserved intestinal mucosa, normal arrangement of crypts and goblet cells, and the absence of inflammatory cells in the colon tissue. Following Neu5Gc intervention, the mucosal lamina propria displayed an inflammatory state characterized by significant infiltration of inflammatory cells. However, after EPF treatment, there was a reduction in inflammatory cell presence and an improvement in pathological symptoms.

In addition, the small intestine tissue structure of the untreated group exhibited a clear and intact morphology characterized by preserved intestinal villi and normal histology. No evident signs of atrophy, necrosis, or infiltration of inflammatory cells were observed. Following Neu5Gc ingestion, the pro-inflammation group displayed intestinal epithelial necrosis, shedding, and an increased presence of inflammatory cells. However, after EPF intervention, a significant reduction in inflammatory cell infiltration was noted, along with the preservation of the integrity of the intestinal epithelium compared to the experimental group. This intervention effectively alleviated the inflammatory condition and played a protective role.

After examining the pathological sections of mouse kidneys and lungs, it was observed that the renal tissues in each treatment group exhibited intact tubular structures and complete nuclei and presented normal glomerular and tubular interstitial states without any significant lesions. Similarly, the alveolar structure of the lung tissue in all groups of mice remained intact, with no signs of edema or inflammatory cell infiltration or bleeding within the alveolar cavity and interstitium. In conclusion, under experimental conditions, short-term red meat Neu5Gc intervention did not induce any organic changes in the kidneys and lungs. However, when compared to liver, small intestine, and large intestine tissues, EPF treatment significantly mitigated tissue damage caused by red meat Neu5Gc while also altering histopathological changes and alleviating its inflammatory progression.

### 3.6. After Undergoing EPF Treatment, the CD4 Cell Count in Neu5Gc-Induced Inflammatory Mice Exhibited a Significant Decrease

The CD4 cell content in mice without Neu5Gc induction was found to be within the normal range (approximately 200), as depicted in Figure 6A,B. However, following Neu5Gc induction, there was a notable increase in CD4 cell content in mice with inflammation, indicating an immune response to inflammation and an elevation of immune cells (around 580). Notably, the group exposed to inflammation promotion exhibited a significant rise in CD4 cells (approximately 780), which differed significantly from the control group (*p* < 0.0001), suggesting a higher level of inflammation and a more robust immune response. Subsequent intervention with EPF resulted in a considerable reduction of CD4 cells in mice; however, their levels remained higher than those observed in the control group (about 230) with statistical significance (*p* < 0.05). These findings indicate that EPF exerts its immunosuppressive function by intervening with CD4 lymphocytes, thereby reducing Neu5Gc-induced inflammation in mice.

### 3.7. EPF Downregulated the Expression of Key Molecules of B Cell Activation in Neu5GC-Induced Inflammatory Mice

The results depicted in Figure 7A–D demonstrate that, when compared to the control group mice, the levels of CD19, CD80, CD86, and CD138 were observed to increase to varying extents in each experimental group of mice. Notably, there was a significant elevation (*p* < 0.0001) in the content of CD molecules within the pro-inflammation group, which correlated with the degree of inflammation present in the body. Following EPF treatment, only the expressions of CD80 and CD86 exhibited a substantial reduction, with a notable difference compared to the pro-inflammation group (*p* < 0.001). Remarkably, EPF effectively downregulated the mRNA expression levels of both CD80 and CD86 without affecting those of CD19 and CD138.

In conjunction with the experimental findings, EPF may exert an impact on signal transmission by modulating the expression of co-stimulatory molecules CD80 and CD86 in the spleen of mice with inflammation, consequently leading to a reduction in B cell activity and anti-Neu5Gc antibody levels.

## 4. Discussion

The presence of high-quality protein, essential amino acids for the human body, B vitamins, and dietary fiber renders meat and its products a highly nutritious food choice. With the evolving lifestyle patterns and increasing emphasis on food quality and safety due to growing demands for health-conscious diets, individuals are not only seeking culinary delicacies but also prioritizing nutritional value. Red meat is abundant in diverse beneficial proteins, vitamins, and trace elements that effectively fulfill our fundamental nutritional requirements [24]. The announcement by the International Organization for Cancer regarding the correlation between the consumption of red meat and its products and the incidence of cancer and other related diseases had a significant global impact [25]. The new meat safety issues caused by the animal-derived hazard Neu5Gc in red meat, however, have not yet been adequately addressed [26].

Currently, various strategies have been explored to effectively mitigate the Neu5Gc content in red meat; however, there are still several limitations, including implementation challenges, high costs, operational difficulties, and compromised food quality [27]. When red meat, which is rich in Neu5Gc, enters the human body, it is recognized as a foreign antigen by the immune system. This recognition triggers the production of heterologous specific antibodies, leading to chronic low-grade inflammation and potentially promoting carcinogenesis. It is evident that antibody production plays a pivotal role in disease initiation. Therefore, reducing the levels of anti-Neu5Gc antibodies in the body can effectively mitigate the risk of inflammation and cancer induction.

Our findings demonstrated that the administration of Neu5Gc to mice resulted in an elevation in the levels of anti-Neu5GC antibodies, accompanied by evident pathological manifestations in the liver, small intestine, and colon tissues, including focal hepatitis, hepatocyte degeneration and necrosis, inflammation of the intestinal mucosa lamina propria, as well as infiltration of inflammatory cells. Prolonged consumption of red meat containing Neu5Gc further exacerbates systemic inflammation and correlates with clinical manifestations involving internal organs such as liver cancer and bowel cancer. Additionally, the ingestion of Neu5Gc significantly increased the levels of inflammatory cytokines in mouse blood. Notably, intervention with EPF exhibited significant differences in reducing various inflammatory markers, thereby confirming its inhibitory effect on Neu5Gc-induced body inflammation.

We investigated the mechanism of EPF using immunohistochemistry and fluorescent quantitative PCR techniques. The findings revealed that early pregnancy factor regulates the immune response dependent on CD4^+^ T-cells while inhibiting the cell-regulated immune response. Following EPF intervention, a significant reduction in CD4 cells was observed in inflammatory mice. Moreover, analysis of key molecules involved in B cell activation demonstrated a notable decrease in the levels of CD80 and CD86 after EPF injection. These co-stimulatory molecules play crucial roles in T cell function by binding to their respective ligands [28]. Under normal conditions, the surface of APC cells does not exhibit any expression of CD80 and CD86. Upon stimulation by foreign antigens, there is a significant increase in the levels of CD80 and CD86 expression, thereby playing a crucial role in co-stimulation. Bretscher proposed that co-stimulatory signals play a pivotal role in T cell activation [29]. In addition, experimental evidence has confirmed that CD80 serves as a regulatory signal for the activation of normal B cells, while an increase in CD86 content enhances B cell activity [30]. After stimulation by Neu5Gc, the body initiates an inflammatory response characterized by the excessive secretion of pro-inflammatory factors such as IL-6, IL-1β, and TNF-α, thereby promoting inflammation. Subsequent intervention with EPF leads to a reduction in B cell activity and the significant down-regulation of CD80 and CD86 expressions. Consequently, it can be inferred that EPF diminishes the expression of CD80 and CD86, inhibits their antigen presentation ability, influences T/B cell activation, disrupts their mediated immune response, and mitigates chronic inflammation induced by Neu5Gc found in red meat.

The progression of chronic inflammation in the context of carcinogenesis and tumors has been extensively investigated. However, there is limited research on the inflammation induced by specific Neu5Gc found in red meat. Our research aims to contribute towards alleviating the inflammatory response caused by Neu5Gc, as well as conducting comprehensive investigations into the signaling pathway and gut microbiota associated with COX-2 [31,32] and NF-κB activation [33], while also exploring its circulatory status in vivo.

## 5. Conclusions

In this study, we analyzed the inhibitory effect of EPF on antibody production and chronic inflammation development in red meat Neu5Gc-induced CMAH^−/−^ C57/BL6 mice, which were successfully generated as a test model simulating humans by knocking out 92 bp on exon 6 of the CMAH gene. The experimental findings suggest that the combination of Neu5Gc in red meat and wild mouse spleen cells can effectively trigger inflammation in experimental mice. After intervention by EPF, the production of Neu5Gc antibodies and the pro-inflammatory factors TNF-α and IL-1β as well as IL-6 and MPO were all reduced. Additionally, there was a significant decrease in CD4 cells in the spleen, and varying degrees of tissue damage in the liver, small intestine, and colon were alleviated. The transcription levels of the second signal activating factors CD80 and CD86 in B cells were inhibited. This further indicates that EPF can inhibit the synthesis of CD4 lymphocytes in the body, interfering with the expression of second signal-activating molecules in B cells, thereby affecting the level of anti-Neu5Gc antibodies in the body. As a result, it weakens inflammation mediated by red meat Neu5Gc and provides a basis for better exploring its pathologic mechanisms. We will use this experiment as a reference in the future to study the data obtained from mouse experiments through research on the human body.

## Figures and Tables

**Figure 1 nutrients-16-00905-f001:**
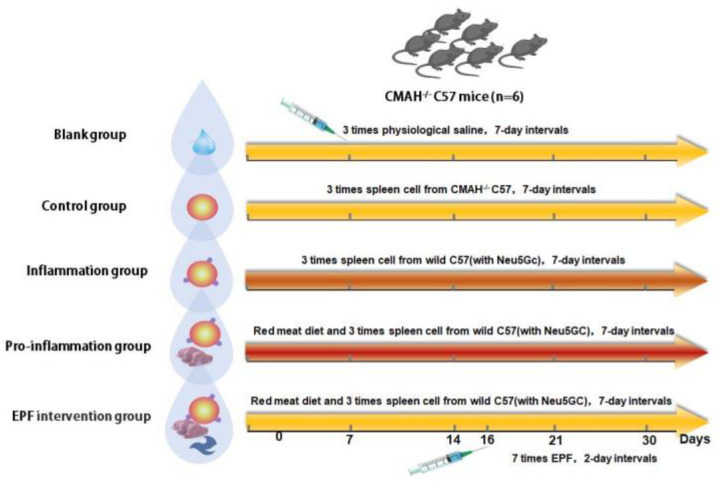
Animal experiment group and implementation plan.

**Figure 2 nutrients-16-00905-f002:**
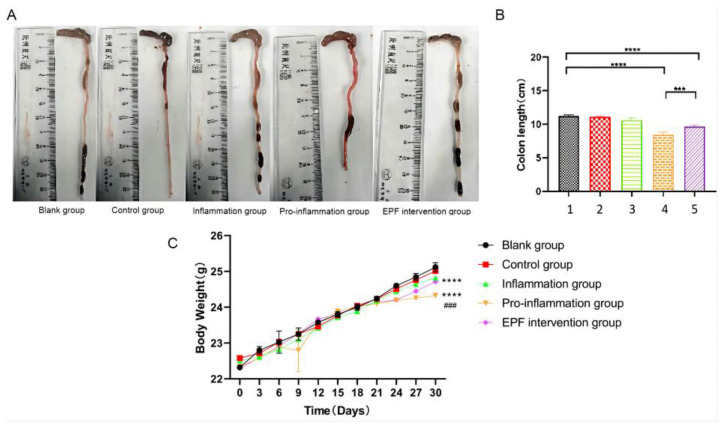
The effect of EPF on colon length and body weight in red meat Neu5Gc-induced inflammation mice. (**A**): effect of EPF on colon length in inflammatory mice. (**B**): statistical results of colon length. (**C**): effect of EPF on Neu5Gc-induced body weight in mice. 1. Blank group 2. Control group 3. Inflammation group 4. Pro-inflammation group 5. EPF intervention group. The data are presented as mean ± SD, and statistical analysis was performed using one-way analysis of variance (ANOVA). Significance levels were indicated as *** *p* < 0.001, **** *p* < 0.0001 and *^###^ p* < 0.001 compared to each respective group.

**Figure 3 nutrients-16-00905-f003:**
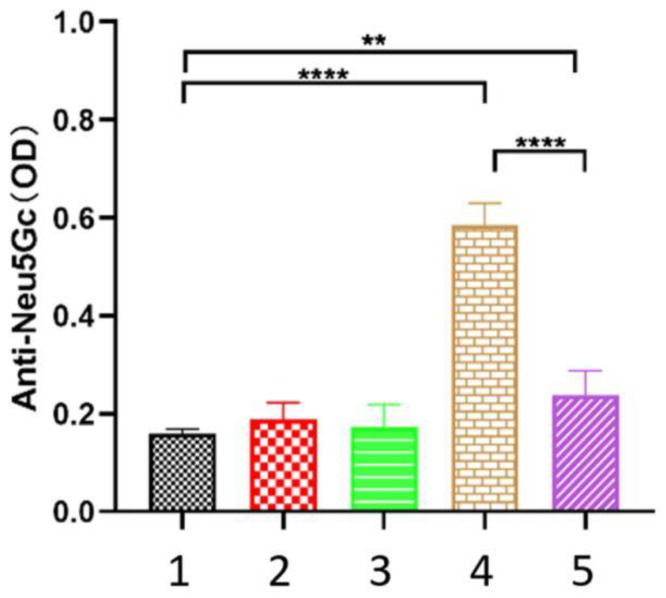
EPF reduced the levels of anti-Neu5Gc antibodies in red meat Neu5Gc-induced inflammation mice. 1. Blank group 2. Control group 3. Inflammation group 4. Pro-inflammation group 5. EPF intervention group. The data are presented as mean ± SD, and statistical analysis was performed using one-way analysis of variance (ANOVA). Significance levels were indicated as ** *p* < 0.01 and **** *p* < 0.0001 compared to each respective group.

**Figure 4 nutrients-16-00905-f004:**
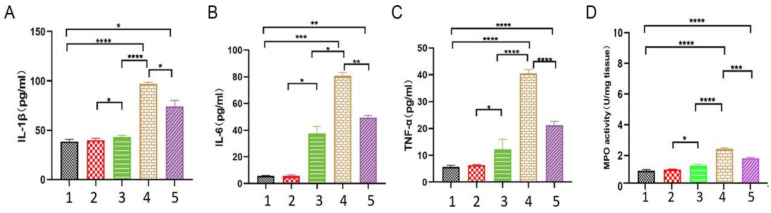
The administration of EPF resulted in a reduction in the levels of inflammatory factors in the serum of mice with inflammation. After blood collection, the serum was separated, and the levels of inflammatory factors in the serum were analyzed. (**A**): the levels of IL-1β; (**B**): the levels of IL-6; (**C**): the levels of TNF-α. (**D**): MPO activity in tissues. 1. Blank group 2. Control group 3. Inflammation group 4. Pro-inflammation group 5. EPF intervention group. The data are presented as mean ± SD and analyzed using one-way analysis of variance. Significance levels were indicated as * *p* < 0.05, ** *p* < 0.01, *** *p* < 0.001 and **** *p* < 0.0001 compared to each respective group.

**Figure 5 nutrients-16-00905-f005:**
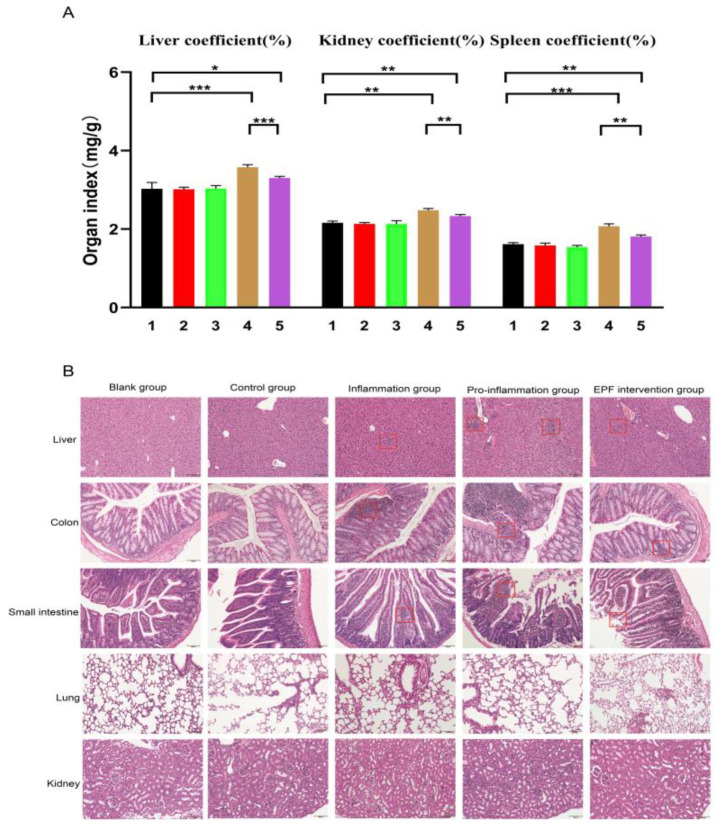
The administration of EPF mitigates histopathological damage in mice with Neu5Gc-induced inflammation. After euthanizing the mice, liver, colon, small intestine, kidney, and lung specimens were collected from each experimental group for subsequent determination of tissue and organ indices through weighing. Subsequently, histopathological analysis was conducted using HE staining to examine the mice in each group. (**A**): effects of EPF on tissue and organ index in Neu5Gc-induced inflammatory mice. 1. Blank group 2. Control group 3. Inflammation group 4. Pro-inflammation group 5. EPF intervention group. (**B**): effects of EPF on histopathology in mice with Neu5Gc-induced inflammation. Significance levels were indicated as * *p* < 0.05, ** *p* < 0.01 and *** *p* < 0.001 compared to each respective group. The red boxes represent the location of the lesion.

**Figure 6 nutrients-16-00905-f006:**
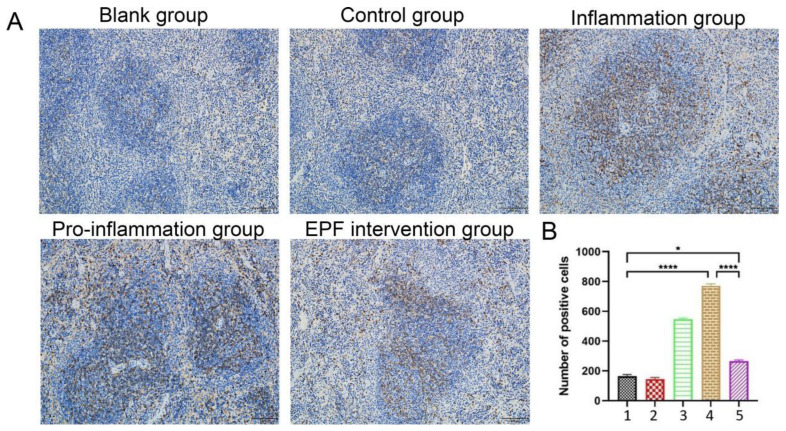
After undergoing EPF treatment, the CD4 cell count in Neu5Gc-induced inflammatory mice exhibited a significant decrease. After euthanizing the mice, the intact spleens were delicately excised and meticulously rinsed with distilled water. Subsequently, the CD4 cell population within the spleens was assessed via immunohistochemistry. (**A**): effect of EPF on the CD4 cell content in the spleen of mice with Neu5Gc-induced inflammation. (**B**): alterations in CD4 cell numbers in the spleen of mice with Neu5Gc-induced inflammation. 1. Blank group 2. Control group 3. Inflammation group 4. Pro-inflammation group 5. EPF intervention group. The data are presented as mean ± SD, and statistical analysis was performed using one-way analysis of variance (ANOVA). Significance levels were indicated as * *p* < 0.05 and **** *p* < 0.0001 compared to each respective group.

**Figure 7 nutrients-16-00905-f007:**
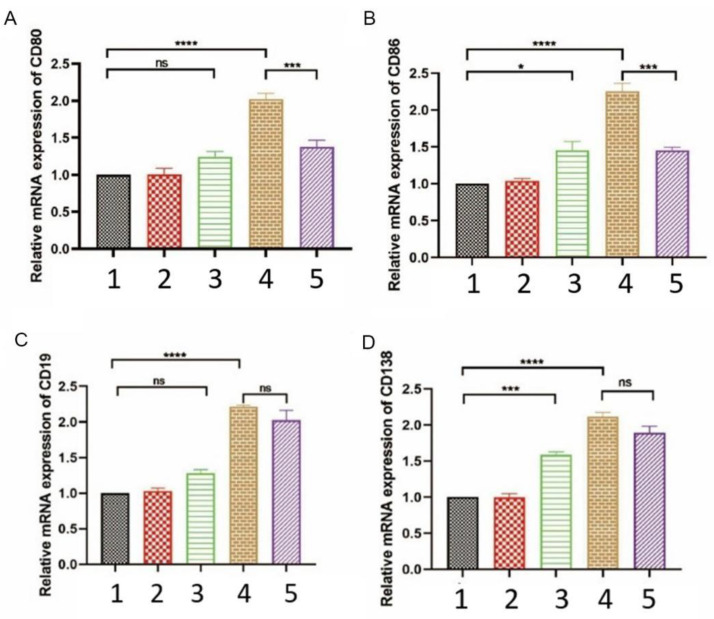
The expression of key molecules involved in B cell activation was downregulated by EPF in Neu5GC-induced inflammatory mice. We assessed the transcriptional levels of CD19, CD80, CD86, and CD138, which are crucial for B cell activation in inflammatory mice, following EPF intervention using fluorescent quantitative PCR. (**A**). Transcriptional status of CD80 mRNA molecule. (**B**). Transcriptional status of CD86 mRNA molecule. (**C**). Transcriptional status of CD19 mRNA molecule. (**D**). Transcriptional status of D138 mRNA molecule. The experimental groups included: 1. Blank group 2. Control group 3. Inflammation group 4. Pro-inflammation group 5. EPF intervention group. The data are presented as mean ± SD, and statistical analysis was performed using one-way analysis of variance (* *p* < 0.05, *** *p* < 0.001 and **** *p* < 0.0001 indicate significance compared to each group). “ns” denotes no significance.

**Table 1 nutrients-16-00905-t001:** Sequences of oligonucleotides used in this study.

Primer’s Name	Sequences (5′–3′)
CD19-F	AGACTGTGTTGTAACCCATCCC
CD19-R	AGGCAAGGGGGGGGTGATGTAAAC
CD80-F	ACCCACAACATAACTGAGTCT
CD80-R	TTCCAACCAAGAGAAGCGGAGG
CD86-F	TAAGCAAGGTCACCGAAA
CD86-R	AGAACACACACACGGTCATATG
CD138-F	ACCAGCAGACACCGAGAC
CD138-R	TGGGAGCCGAGTCTCATGG
β-action F	GGCTGTATTCCCCTCCATCG
β-actin-R	CCAGTTGGTAACAATGCCATGT

**Table 2 nutrients-16-00905-t002:** Rate of rosette formation.

Group	Number	Lymphocyte Count	Rosette Forming Rate			Average Value
Control	3	200	34%	40%	38%	37.3%
EPF	3	200	20%	17%	21%	19.3%
EPF + Antibody	3	200	30%	31%	34%	31.7%

## Data Availability

The original contributions presented in the study are included in the article, further inquiries can be directed to the corresponding author.

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
