# Peer review of "The Inhibitory Effect of Early Pregnancy Factor on Red Meat Neu5Gc-Mediated Antibody Production in CMAH−/− Mice"

_nutrients, 2024, doi:10.3390/nu16060905_

Round 1

Reviewer 1 Report

Comments and Suggestions for Authors

The presented passage delves into the impact of red meat consumption on the immune system, particularly focusing on the potential risks associated with the presence of N-Glycolylneuraminic Acid (Neu5Gc). The study introduces the intriguing concept of utilizing early pregnancy factor (EPF) to regulate immune responses, using a CMAH gene knockout in C57 mice as an animal model simulating human conditions.

The findings of this research are notable, as they highlight the inhibitory effect of EPF on antibody production and chronic inflammation in CMAH-/- C57 mice induced with Neu5Gc from red meat. The observed outcomes include the alleviation of slow weight gain and colon length shortening, indicating a positive impact of EPF intervention. Furthermore, EPF treatment demonstrated a significant reduction in anti Neu5Gc antibodies, as well as a decrease in inflammatory factors such as IL-6, IL-1β, TNF-α, and MPO activity. The protective effects extended to liver and intestinal tissues, with reduced damage observed. Additionally, EPF treatment showed promise in diminishing CD4 cell content and the expression of B cell activation molecules CD80 and CD86 in mice.

Comments:

The study relies on CMAH-/- C57 mice as an experimental model simulating humans. While mice are often used in research, their biological differences from humans might not fully represent the complexity of human responses to red meat consumption and EPF intervention. While the findings are promising, it is crucial to acknowledge the need for further human-centric studies to validate and extrapolate these results to broader health implications.

Author Response

nutrients-2877890
Response to reviewers
Thank you for your letter and comments concerning our manuscript entitled “The inhibitory effect of Early Pregnancy Factor on red meat Neu5Gc-mediated antibody production in CMAH-/- mice” nutrients-2877890). These comments are very valuable and helpful for revising and improving our paper. Below are our responses to the editor’s comments. In addition, we have found that this article will be published in the special issue of Nutrition and Public Health. May I ask if it is possible to publish this article in the regular issue? Looking forward to your reply.
In order to facilitate the review, the red font indicates the modification in the revised manuscript. The responds to the editor’s comments are as follows:
Editor comments:
1. The study relies on CMAH-/- C57 mice as an experimental model simulating humans. While mice are often used in research, their biological differences from humans might not fully represent the complexity of human responses to red meat consumption and EPF intervention. While the findings are promising, it is crucial to acknowledge the need for further human-centric studies to validate and extrapolate these results to broader health implications.
Response: Dear reviewer, thank you for your suggestion. We understand what you mean. Putting people at the center is the ultimate goal of our experiment, and we will continue to focus on this direction for further research.

Reviewer 2 Report

Comments and Suggestions for Authors

The work is very interesting but it is not clear to me why the authors did not use a control group that received intraperitoneal injections of spleen cells from CMAH-/- C57 mice and fed with a diet rich in red meat. The authors should add this check or justify why they didn't.

Minor points:

Abstract: rewrite the abstract more clearly and explain why the CMAH-/- mouse strain is used and what the gene CMAH is important.

Lines 46-59: this part is unclear and needs to be rewritten

Lines 81-84: explain this strain well, adding references where it is used and the effects that are observed

Line 89: are you sure that the strain is C57 and not c57/BL6?

Line 168:  Is this reference correct?

If it is possible to write the groups in the text in the exact same way as they were shown in figure 1

Figure 1: replace READ with RED

Author Response

nutrients-2877890
Response to reviewers
Thank you for your letter and comments concerning our manuscript entitled “The inhibitory effect of Early Pregnancy Factor on red meat Neu5Gc-mediated antibody production in CMAH-/- mice” nutrients-2877890). These comments are very valuable and helpful for revising and improving our paper. Below are our responses to the editor’s comments. In addition, we have found that this article will be published in the special issue of Nutrition and Public Health. May I ask if it is possible to publish this article in the regular issue? Looking forward to your reply.
In order to facilitate the review, the red font indicates the modification in the revised manuscript. The responds to the editor’s comments are as follows:
Editor comments:
1. The work is very interesting but it is not clear to me why the authors did not use a control group that received intraperitoneal injections of spleen cells from CMAH-/- C57 mice and fed with a diet rich in red meat. The authors should add this check or justify why they didn't.
Response:Thank you for your suggestion. In our preliminary experiment, we fed red meat and injected CMAH-/- mice with cells without Neu5Gc. The experimental results showed no difference compared to the blank group. Therefore, in order to retain the single variable of the control group and the Inflammation group, which is the injection of spleen cells with/without Neu5Gc, we chose to simplify the control group by not feeding red meat and injecting spleen cells without Neu5Gc.

2. Abstract: rewrite the abstract more clearly and explain why the CMAH-/- mouse strain is used and what the gene CMAH is important.
Response:Thank you for your suggestion. The abstract of the manuscript has been rewritten, highlighting the reasons for using CMAH-/- mice and the significance of the CMAH gene (line 12-28).

3. Lines 46-59: this part is unclear and needs to be rewritten.
Response:Thank you. I have carefully rewritten this section, please review it (line 45-57).

4. Lines 81-84: explain this strain well, adding references where it is used and the effects that are observed.
Response:Thank you. I have added references based on experimental observations and provided explanations for their effects.

5. Line 89: are you sure that the strain is C57 and not c57/BL6?
Response:Thank you. I have changed C57 to c57/BL6 (line 91).

6. Line 168:  Is this reference correct?
Response:Thank you. I have replaced the reference with a new one (line 170).

7. If it is possible to write the groups in the text in the exact same way as they were shown in figure 1
Response:Thank you. I have modified the text as shown in Figure 1.

8. Figure 1: replace READ with RED.
Response:Thank you. I have corrected the issue in Figure 1 (line 131).

Round 2

Reviewer 2 Report

Comments and Suggestions for Authors

Sorry, but also in this version, in the abstract the author not specify what
the CMAH gene encodes and why it is important for the study.
Moreover, this sentence '
In our preliminary experiment, we fed red meat and injected CMAH-/- mice with cells without Neu5Gc. The experimental results showed no difference compared to the blank group. Therefore, in order to retain the single variable of the control group and the Inflammation group, which is the injection of spleen cells with/without Neu5Gc, we chose to simplify the control group by not feeding red meat and injecting spleen cells without Neu5Gc.' should be inserted into the text.

Finally the strain is always C57/Bl6 and not C57, even in the abstract

Author Response

nutrients-2877890

Response to reviewers

Thank you for your letter and comments on our article titled "Inhibition of Red Meat Neu5Gc Mediated Antibody Production by Early Pregnancy Factors in CMAH -/- Mice" (Nutrition-2877890). These opinions are very meaningful and valuable to us. Here is our response to your feedback.

In order to facilitate the review, the red font indicates the modification in the revised manuscript. The responds to the editor’s comments are as follows:

Editor comments:

1.Sorry, but also in this version, in the abstract the author not specify what the CMAH gene encodes and why it is important for the study.

Response:Thank you, editor. I have added the content of the CMAH gene and its importance to our experiment in the abstract(line17-23).

2.Moreover, this sentence 'In our preliminary experiment, we fed red meat and injected CMAH-/- mice with cells without Neu5Gc. The experimental results showed no difference compared to the blank group. Therefore, in order to retain the single variable of the control group and the Inflammation group, which is the injection of spleen cells with/without Neu5Gc, we chose to simplify the control group by not feeding red meat and injecting spleen cells without Neu5Gc.' should be inserted into the text.

Response:Thank you for your suggestion. I have added this paragraph to the article(line125-130).

3.Finally the strain is always C57/Bl6 and not C57, even in the abstract

Response:Thank you for your suggestion. I have changed it from C57 to C57/Bl6.